# The Role of Cancer Stem Cells and Mechanical Forces in Ovarian Cancer Metastasis

**DOI:** 10.3390/cancers11071008

**Published:** 2019-07-18

**Authors:** Michael E. Bregenzer, Eric N. Horst, Pooja Mehta, Caymen M. Novak, Taylor Repetto, Geeta Mehta

**Affiliations:** 1Department of Biomedical Engineering; University of Michigan, Ann Arbor, MI 48109, USA; 2Department of Materials Science and Engineering; University of Michigan, Ann Arbor, MI 48109, USA; 3Macromolecular Science and Engineering; University of Michigan, Ann Arbor, MI 48109, USA; 4Rogel Cancer Center, University of Michigan, Ann Arbor, MI 48109, USA

**Keywords:** ovarian cancer metastasis, cancer stem cells, mechanical stimulation, mechanotransduction, tension, compression, shear stress, viscoelastic, ascites, matrix stiffness

## Abstract

Ovarian cancer is an extremely lethal gynecologic disease; with the high-grade serous subtype predominantly associated with poor survival rates. Lack of early diagnostic biomarkers and prevalence of post-treatment recurrence, present substantial challenges in treating ovarian cancers. These cancers are also characterized by a high degree of heterogeneity and protracted metastasis, further complicating treatment. Within the ovarian tumor microenvironment, cancer stem-like cells and mechanical stimuli are two underappreciated key elements that play a crucial role in facilitating these outcomes. In this review article, we highlight their roles in modulating ovarian cancer metastasis. Specifically, we outline the clinical relevance of cancer stem-like cells, and challenges associated with their identification and characterization and summarize the ways in which they modulate ovarian cancer metastasis. Further, we review the mechanical cues in the ovarian tumor microenvironment, including, tension, shear, compression and matrix stiffness, that influence (cancer stem-like cells and) metastasis in ovarian cancers. Lastly, we outline the challenges associated with probing these important modulators of ovarian cancer metastasis and provide suggestions for incorporating these cues in basic biology and translational research focused on metastasis. We conclude that future studies on ovarian cancer metastasis will benefit from the careful consideration of mechanical stimuli and cancer stem cells, ultimately allowing for the development of more effective therapies.

## 1. Introduction

Ovarian cancers comprise a heterogeneous collection of neoplasms with distinct clinicopathological and molecular features, and prognoses that arise from, or involve the ovary. They are the 5th leading cause of cancer deaths among women in the developed world and the most lethal gynecological malignancy [1]. Late diagnoses and an approximately 80% relapse rate after successful primary therapy, are largely responsible for this dreary statistic [2]. The late diagnosis is especially problematic considering that the 5-year survival rate drops from 45% to 25% when diagnosed in advanced stages of tumor progression [3]. In fact, 70% of ovarian cancer diagnoses occur when the tumor has already spread in the abdominal cavity [3,4] indicating stage III disease and worse prognosis [5].

According to the histotypes, ovarian carcinomas are classified into several morphological categories, including: serous, mucinous, endometrioid, and clear-cell carcinomas, transitional-cell Brenner tumors, mixed, and undifferentiated [6]. The most common and malignant form of ovarian carcinoma is high-grade serous, which presents in an advanced stage and is an inherently aggressive malignancy, thus accounting for the majority of ovarian cancer deaths [7]. High-grade serous carcinomas lack defined architecture and present as sheets of malignant cells, with nuclei that are often enlarged and dysmorphic. They are also characterized by genetic alterations in *TP53*, *BRCA1*, *BRCA2*, *PTEN,* expression of *WT1*, *ERα*, and *PAX8*, and associated effects on DNA repair that lead to genomic instability and high copy number variability [8,9,10,11]. Although there has been no clinical or diagnostic application yet, gene expression sets have segregated high-grade serous carcinoma into four descriptive groups: proliferative, mesenchymal, immune, and differentiated [8,12]. The metastasis of high-grade serous carcinomas often involves fallopian tubes, ovarian surfaces, peritoneal surfaces, and the omentum, and is highly lethal in nature [7].

The often-conflicting notions on the origin of ovarian cancers can be attributed to the fact that cells in the ovarian tumor have little to no phenotypic resemblance to the cells in the ovary [13]. It is interesting to note that the many cellular subtypes of ovarian cancer have their origins outside of the ovary. As an example, the fallopian tube fimbria or ovarian cortical inclusion cysts are thought to be the origin of differentiation of high-grade serous carcinoma from undifferentiated cells. Concordantly, the formation of serous tubal intraepithelial carcinoma (STIC) in the distal fallopian tube epithelium is often an indicator for high-grade serous ovarian carcinoma [7,13]. Meanwhile, low-grade serous carcinoma, which shows phenotypic similarity to high-grade serous carcinomas, but differs in molecular pathways, arise from endosalpingiosis or papillary tubal hyperplasia and have a serous borderline tumor as the precursor lesion [13]. The extremely high heterogeneity in origin, morphology, molecular and immunohistochemical signature, across the various ovarian cancer subtypes and within a single tumor, represents a major challenge in understanding the evolution and biology of ovarian cancers, and also is one of the major causes of treatment failure [6,13].

## 2. Metastasis in Ovarian Cancers

The metastatic spread of the primary tumor to secondary locations causes approximately 90% of all cancers to become fatal. Therefore, understanding of metastatic processes, metastatic cell phenotypes, and metastasis promoting characteristics of the tumor microenvironment (TME) is crucial to improving clinical outcomes. For this reason, metastasis is widely studied in basic and translational medicine [4,14]. Here, we review metastasis in ovarian cancer and its modulation by cancer stem-like cells (CSCs) and mechanical forces in the TME.

In ovarian cancers, metastasis can occur through hematogenous, lymphatic, or transcoelemic routes, with transcoelemic being the most common [15]. Hematological metastasis generally requires four steps: (1) local tumor cell invasion; (2) intravasation into the vasculature; (3) extravasation out of the vasculature; (4) and colonization at a secondary location [16]. This particular form of metastasis is less common in ovarian cancer at the time of diagnosis [15] leading to doubts regarding the ability of ovarian cancer to spread through the vasculature [17]. However, recent work has shown that ovarian cancer cells are capable of hematogenous metastasis, using a parabiosis model to demonstrate that hematogenous metastasis is driven by ErbB3-Neuroegulin1 signaling, and is a key contributor to the high percentages of omental metastasis observed in ovarian cancer [4,17,18]. In particular, Coffman et al. used an intravenous injection of ovarian tumor cells, a murine subcutaneous tumor model, and a human subcutaneous tumor model to show the capacity of ovarian tumor cells to metastasize in the vasculature [17]. Finally, hematological metastasis has also been linked to lymphatic metastasis, which can serve as a milestone between metastatic ovarian cancer cells in the ascites and the vasculature [15]. Despite these findings, the lack of research into the mechanism of hematogenous metastasis necessitates further studies to better understand the contribution of this mode of metastasis to overall metastatic burden in ovarian cancers.

Aside from migration through the vasculature, ovarian cancer is also known to metastasize directly in surrounding organs, through the malignant ascites fluid, or through the lymphatic system [4]. In serous ovarian cancer, lymphatic spread is most common to the para-aortic region, particularly above the inferior mesenteric artery, while in non-serous tumors, para-aortic metastasis occurs with approximately equal frequency as pelvic node metastasis [19]. Advanced cases of ovarian cancers tend to have more incidences of nodal metastasis, which is associated with worse prognosis, especially when patients also have peritoneal metastasis [20]. This corresponds with the association of both lymph node metastasis and buildup of ascites fluid with poor clinical outcomes [20,21]. In fact, the ascites fluid facilitates entry of cancer cells into the lymphatic system through the main drainage channels for fluid absorption in the peritoneal cavity, which reside in the diaphragm. Through the lymphatic system, ovarian cancer cells can also enter the vasculature via the left subclavian vein, perhaps contributing to the presence of ovarian cancer cells in the bloodstream at diagnosis [15,22,23]. Metastasis to the lymphatic system can then result in further ascites build-up as a result of lymphatic vessels blocked by cancer cells [24].

The build-up of ascites fluid is added to by the increased presence of VEGF, which promotes leaky vasculature, resulting in a large volume of fluid through which transcoelemic metastasis occurs [25]. In this process, metastatic ovarian cancer cells undergo epithelial-to-mesenchymal transition (EMT) and enter the ascites fluid by shedding from the primary tumor as single cells or in groups as spheroids and spread passively through the turbulent movement of the peritoneal fluid [4,25,26]. Metastatic ovarian cancer cells then implant into the mesothelium, and rarely invade deeper into the peritoneum than the mesothelial lining [4,26]. Once a colony is established at a secondary site, the cells can undergo a mesenchymal-to-epithelial transition (MET) and begin to grow rapidly [4]. For a more detailed description of metastatic processes in ovarian cancer, readers are referred to the following references [4,25,27,28].

While ovarian cancer research has come a long way towards understanding the mechanisms of metastasis in ovarian cancer, many details are yet unknown and will require a significant amount of research to develop effective therapeutics that can prevent and treat metastatic disease. CSCs and mechanical stimulation are two crucial elements of the ovarian cancer microenvironment that influence metastasis and could serve as foundations for metastasis targeting interventions. In the next sections of this review, we focus on the roles that cancer stem-like cells and mechanical stimulation play in metastasis of ovarian cancers and discuss challenges facing future metastasis research with respect to these cells and mechanical stimulation.

## 3. The Roles of Ovarian Cancer Stem-Like Cells (CSCs) in Metastasis

Cancer stem-like cells (CSCs) are a rare population of stem-like cells that have the capacity to self-renew, differentiate through asymmetric cell division, and resist anoikis (Figure 1B). CSCs are capable of generating an entire tumor and tend to be more chemo- and radioresistant due to their quiescent state and upregulation of efflux pumps [3,29]. CSCs are also highly plastic, can evade the immune system, have efficient DNA damage repair mechanisms, and can easily adapt to hostile environments [29]. Consequently, CSCs have been implicated in tumor recurrence, chemoresistance, metastasis, and thus poor clinical outcomes [3]. These malignant characteristics of CSCs highlight the need for the development of targeted therapies to inhibit their functionality or sensitize them to traditional therapies and thus improve clinical outcomes [29]. Here we discuss various ways in which ovarian CSCs can be defined and characterized and the role they play in metastasis.

Despite the clear importance of CSCs in clinical prognosis and their close association with metastasis, there remains some debate regarding markers that define CSC populations in ovarian cancers [3]. CD44, CD133, CD24, CD117, Nestin, Nanog, and Oct3/4, as well as functional markers like ALDH1A1 and ABC transporters, have all been reported to be ovarian CSC markers, and indicators of chemoresistance, tumor-initiating potential, sphere-forming ability, invasiveness and poor prognosis [3,4,30,31]. However, these markers are not necessarily expressed together, and may not be indicative of the entire repertoire of CSC populations, as there have also been conflicting reports of stemness with each of these markers.

For example, CD24+ cells have been shown to self-renew, to differentiate, to resist chemotherapies, and to play a role in migration and metastasis in some studies [30,31], while others report similar characteristics for CD24− cell populations [30,32,33]. Furthermore, dual expression of more than one of these markers, such as CD133 and ALDH1A1 has been reported to indicate increased stem-like properties than cells with single CSC marker expression [3,34]. These data suggest that cells with more than one CSC marker may be further upstream of more differentiated cells that express only one CSC marker. Contrarily, reports of increased stemness in CD44+/CD24− populations compared to CD44+/CD24+ populations [3] indicate a non-linear relationship between expression of multiple markers and stemness and emphasizes the importance of using functional studies to identify CSC populations. The discrepancy may be due to heterogeneity between patients, different experimental methods, and the importance of specific isoforms [3,35]. The variation observed in the expression of these CSC markers also suggests the existence of multiple CSC subsets, complicating the development of future therapeutics and the use of these markers as prognostic indicators. For a more detailed review of different CSC markers, readers are referred to the following references [3,30,36]. The plasticity of CSCs, which can differentiate and transdifferentiate into different cell lineages, further obscures identification of definitive CSC populations [37,38].

A prime example of CSC plasticity is their capacity to undergo EMT and MET, which links them to metastasis by allowing them to migrate to distant locations and then initiate proliferation once established in secondary locations [4,35,39,40], whether through the ascites, the lymphatic system, or the vasculature. In fact, it has been shown that the process of EMT itself can generate cells with stem cell characteristics and that disruption of the EMT process can result in decreased expression of CSC markers CD117+/CD44+ [28,41]. Furthermore, cells that undergo EMT are generally more chemoresistant, similar to CSCs [28]. Additionally, the process of EMT can be triggered by treatment with cisplatin, a standard chemotherapy treatment for ovarian cancer, which is revelatory in the finding that cisplatin can also enrich for ovarian CSCs [28,42]. Therefore, the link between ovarian CSCs, EMT and the tumor relapse and chemoresistance found in ovarian cancer patients following primary treatment, is established [2,3].

In ovarian cancer, CSCs also play a unique role in metastasis given their ability to resist anoikis, which allows them to survive in non-adherent conditions within the lymphatic and vascular systems, as well as, the ascites fluid within the peritoneal cavity [3,43]. This survival advantage in non-adherent conditions ultimately allows metastasizing stem like cancer cells to eventually adhere to secondary locations and generate secondary tumors via the tumorigenicity associated with their stemness. As CSCs are closely associated with ovarian cancer EMT as discussed earlier, and are upregulated in the ascites, it follows that the ascites fluid may be a contributing factor in enhanced chemoresistance, stemness, and metastasis in ovarian cancer. This shift in phenotype may be due in part to mechanotransduction, however, future studies are critically needed to bridge the large gap in our understanding of these phenomena. As a whole, the significance of various CSC subsets in the deadly process of metastasis remains unclear illuminating the need for more rigorous studies of metastasis with a focus on ovarian CSCs. Specifically, to address this gap in knowledge, more robust CSC research needs to be conducted to identify potential differences in functionality between different CSC subsets and understand patterns of CSC differentiation, thereby facilitating the development of CSC targeting metastasis therapies.

## 4. Interactions between Ovarian Cancer Stem-Like Cells and Ovarian Tumor Microenvironment that Modulate Metastasis

Ovarian cancer cells and the stroma interact bidirectionally, modulating the ascitic fluid contents and promoting pro-tumoral phenotypes within the stromal cells and regulating processes to favor metastasis [44,45,46] (Figure 1B). The reciprocal interaction between the stroma and tumor cells, with the help of release of cytokines and soluble factors, results in the generation of specific niches within the tumor and more importantly, aids in the programming of cancer cells to CSCs. This makes the tumor stroma an irrefutable puzzle piece, being studied for its significance in the initiation, progression, metastasis, recurrence, chemoresistance, and invasiveness of the tumor. Not surprisingly, the tumor stroma is also being studied as a target of antitumor therapy and for its potential role as a diagnostic marker. Extensive details of tumor microenvironment elements have been reviewed elsewhere [47]; in this manuscript, we focus on the effects of tumor-stroma interactions and its role in metastasis. Briefly, tumors are considered to be akin to complex organs, with heterogeneous cell populations including adipocytes, endothelial cells, and mesenchymal cells such as cancer associated fibroblasts (CAFs) and mesenchymal stem cells (MSCs), along with cells from the immune system such as tumor-associated macrophages (TAMs), regulatory T cells and cytotoxic T cells, among other cell types. Through reciprocal interactions with the heterogenous supporting cells, tumor cells avoid apoptosis, proliferate, invade and metastasize. In this way, tumors behave like abnormal organs, with dysregulated structure and function compared to homeostatic organs [47,48,49].

The role of the non-cancer cells found in the TME is exemplified by CAFs, which arise from activated fibroblasts and differentiated mesenchymal stem cells [50]. This activation is often in response to environmental factors like inflammation and hypoxia, which leads to an inflammatory TME, cancer metastasis and increased invasiveness [51,52,53,54,55,56]. Mesothelial cells, the foremost barrier faced by metastatic ovarian cancer cells, have also been shown to convert to myofibroblasts through mesothelial to mesenchymal transition (MMT) [57]. Secretory factors like TGF-β1, inflammatory cytokines and chemokines, reactive oxygen species (ROS) as well as matrix metalloproteinases (MMPs) also activate CAFs in response to signaling from ovarian cancer cells [53]. In ovarian cancer, CAFs promote cancer cell proliferation by excessive secretion of hepatocyte growth factor (HGF), through the activation of cMet/PI3K/Akt pathways and glucose-regulated protein 78 (GRP78) that promote chemoresistance, cancer cell invasion and cell migration [58,59]. Tumor progression and a pro-inflammatory TME that is conducive to metastasis is the result of activation of NF-κB and JNK signaling pathways, which are mediated by versican binding to CD44. Overexpression of versican, in turn, is achieved by crosstalk between cancer cells and CAFs, which is regulated by TGF-β/TGF-β receptors and the SMAD pathway [55]. CAFs also secrete VEGF-C which promotes angiogenesis, tumor progression and chemoresistance through their effect on endothelial cells [60]. Through the secretion of MMPs, extracellular matrix (ECM) components, and enzymes, CAFs also promote TME remodeling [61], which may influence the mechanical stimuli in the TME. Altered mechanical stimuli, has the potential to affect CSC fate as discussed later. Once transformed, CAFs can also recruit lymphocytes that produce inflammatory signals in the TME to aid cancer progression and can also suppress cytotoxic T lymphocyte function, thereby protecting cancer cells. 

Carcinoma-associated MSCs (CA-MSCs) are another stromal cell population that inhabit the majority of ovarian TMEs. Lacking in tumorigenic potential, CA-MSCs differentiate into CAFs and adipocytes, thus contributing to the changing TME [62]. CA-MSCs also increase the population of ovarian CSCs through upregulation of TGF-β/BMP family members [63]. Higher expression of TGF-β super family members in CA-MSCs also contribute to their involvement in tumor growth promotion, enrichment of CSCs and chemotherapeutic resistance [62]. Finally, MSCs also promote angiogenesis by inducing VEGF and HIF1α expression in ovarian cancer thereby further contributing to tumor progression and CSC phenotypes [38,64].

In the ovarian TME, MSCs as well as tumor cells, have been shown to recruit macrophages, which in turn secrete ECM components and enzymes resulting in ECM reorganization and remodeling [65], again likely influencing mechanical stimuli sensed by cancer cells. In the presence of specific factors like IL-4, IL-10, and IL-13 these macrophages become polarized and differentiate into an M2-like phenotype [66], also called tumor-associated macrophages (TAMs) or alternatively activated macrophages (AAMs), and secrete IL-4, IL-5, and IL-6 [67,68]. Multiple mechanisms including augmented c-Jun and NF-κB activity facilitate ovarian cancer cell invasion by TAMs [69]. Similar to CAFs and CA-MSCs, the TAMs and other immune cells in the inflammatory microenvironment can also promote ovarian CSCs through secretion of cytokines, such as IL-17, which can activate CSC promoting pathways like NF-κB and p38-mitogen activated protein kinase (MAPK), which promote CSC self-renewal and progression of ovarian cancer [70].

The different cell types, secretome, proteome and signaling enable the formation of a metastatic niche for the tumor cells for self-renewal, tumor initiation, and long-term survival. The bidirectional interaction between ovarian CSCs and the tumor microenvironment, allows CSCs to maintain stemness and differentiate into other cell types to support tumor progression depending on the CSC niche. Certain conditions like hypoxia, inflammation, and increased angiogenesis, mentioned in the next section, allow the tumors to have a favorable environment to grow and metastasize, bypassing the checks and balances that maintain homeostasis.

In order to grow and metastasize, tumors need to overcome the anti-tumor immune responses, which are facilitated by myeloid-derived suppressor cells (MDSCs), macrophages, and Treg cells within the pre-metastatic niche [71,72]. To do this, tumor cells hijack the adaptive and innate immune responses in order to avoid detection and destruction, leading to immunosuppression, cancer cell proliferation and metastasis [73]. Natural killer (NK) cells and CD8^+^ T cells attack cancer cells in early stages of tumorigenesis but eventually the tumor and its metastatic derivatives establish an immunosuppressive pre-metastatic niche [72]. Polarized M2 macrophages or TAMs contribute to this immunosuppressive environment, as well as, angiogenesis and matrix remodeling to support tumor growth in contrast to classically activated or M1 macrophages which inhibit tumor progression [44]. MDSCs similarly cause immunosuppression and disrupt immunosurveillance by interfering with T cell activation, and inhibition of NK cell cytotoxicity [74]. While Treg cells maintain homeostasis in physiologically normal conditions, they play a different role in the ovarian tumor microenvironment, ultimately resulting in suppression of the immune system [75]. Immune cells further contribute to metastasis by the production of pro-inflammatory factors, such as TGF-β, TNF-α, IL-6, IL-1, along with hypoxia, which can regulate the expression of transcription factors and proteases, thus inducing EMT through NF-κB and STAT3 activation [76] and propelling metastasis.

Furthermore, elevated levels of VEGF in ovarian cancer, produced not only by cancer cells but TAMs, CAFs, and cancer-associated adipocytes (CAAs) leads to increased angiogenesis and endothelial cell survival, proliferation, migration and increased vascular permeability [77,78]. VEGF induces the expression of CXCL12 receptor which pairs with the increased secretion of CXCL12 from cancer cells in hypoxic TMEs resulting in angiogenesis [79]. Peritoneal carcinomatosis associated with malignant ascites formation is promoted by elevated VEGF [78,80]. Additionally, the adipose rich omentum and peritoneal surfaces are a favorable site for metastasis for ovarian tumor cells, due to the proximity and an inherent tropism [4,81]. This tropism is thought to be due in part to adipocyte interactions with cancer cells through IL-8 [82]. Finally, studies also illustrate that tumors can release exosomes, which play a significant role in intracellular communication and promote tumor progression. They are directly or indirectly involved in changing the TME as they could be taken up by NKs and B cells [83,84,85]. The examples described in this section exhibit the various CSC niche dependent bidirectional interactions between ovarian CSCs and the TME elements. These interactions allow for the maintenance and enrichment of stemness traits to support ovarian tumor progression and metastasis. Importantly, these interactions are not independent of the mechanical environment, as recruitment of host cells and remodeling of the TME influence the mechanical forces experienced by the cells.

## 5. Nexus between Cancer Stem Like Cell Phenotypes and Mechanotransduction

While there are few studies in ovarian cancer linking mechanical stimuli to the increased CSC phenotypes, mechanotransduction research in the context of other cancers provides reason to speculate that a connection between ovarian CSCs and mechanical stimuli exists. An example within breast cancer research proposed by Triantafillu et al. in 2017 highlighted the increased expression of EMT markers in non-adherent culture [86]. Additionally, when the non-adherent breast cancer cells were subjected to fluid shear stress they displayed an increase in cancer stem like cell phenotype [86,87]. As CSCs are enriched within the malignant ascites of ovarian cancer patients, and the ascites fluid inevitably produces shear forces, it follows that this fluid-induced shear stress may be an important factor in CSC enrichment within the ascites. In fact, this connection has been supported in ovarian cancer [88,89,90] and has implications in metastasis, chemoresistance, and recurrence.

More generally, it has also been suggested that CSC phenotypes are linked to mechanotransduction through mechanosensitive transcription factors, YAP/TAZ. These transcription factors intertwine with Wnt and TGF-β pathways, which in turn are associated with stemness [91]. Signaling pathways involving YAP/TAZ, as well as pathways like NF-κB, ERK, Rho/ROCK, and FAK can all be triggered by mechanosensing of ECM proteins or mechanical stimuli like stiffness and stretch to promote CSC characteristics including self-renewal, survival, drug resistance, proliferation, migration, and invasion [91,92,93]. While this connection between mechanical environment and CSC phenotypes is often overlooked in ovarian cancer research, it is critical in understanding both how CSCs are regulated and how metastasis progresses. Having discussed the role of cancer stem-like cells in metastasis and the interplay of mechanical stimuli and CSCs in ovarian cancer, the next part of this review will take a more detailed look at the role of mechanical stimuli in ovarian cancer metastasis.

## 6. Mechanical Stimuli in Ovarian Tumor Microenvironment and Metastasis

Within the ovarian tumor microenvironment cancer cells experience a wide range of mechanical stimuli. These include: (1) the displacement of the native environment from the ever-expanding tumor, (2) the ECM remodeling from the desmoplastic response, (3) the development of ascites within the peritoneal cavity, and, (4) the dynamic movements from everyday living (Figure 1A,B). These forces contribute not only to the progression of the disease via cellular signaling changes known as mechanotransduction but also through the facilitation of cellular aggregates to new points of metastasis [15]. In the last few years, a new narrative has emerged implicating mechanotransduction as both a driving force of metastasis and a factor that may be promoting cancer stem cell-like populations. Due to the ovary’s anatomic location, ovarian cancer cells are exposed to the peritoneal microenvironment. In the pathologic state, this environment is filled with ascites fluid under constant hydrostatic pressure. The peritoneal cavity is continuously changing shape, accounting for movements of surrounding organs, musculoskeletal dynamics, and gravity. The associated fluid movement stimulates cells with high levels of shear stress. Dissemination to other parts of the body and advancement in ovarian cancer stages largely occurs due to the isolation of the ovarian cancer cells in the peritoneal cavity. Due to the unique microenvironmental niches for ovarian cancers, the tumor cells experience the following mechanical stimuli: shear stress, compressive stress, tensile stress, matrix stiffness, and stress relaxation. These stresses not only build over time but also accompany the progression of the disease. This concomitantly exposes cancer cells to chronic aberrant mechanical stimuli and aids in ovarian cancer progression, forming a positive feedback loop. The impact of these mechanical stimuli on ovarian cancer progression and metastasis has only begun to be studied and understood. Mechanotransduction studies isolating these stimuli have begun to show the importance of these factors in assisting in the metastasis and progression of a variety of cancers. Additionally, few studies have started to link mechanical stimulus with the expression of CSC markers [94] emphasizing the importance of continued exploration of these relations. The processes, by which the many mechanical forces and their consecutive actions directly affect ovarian cancer and the pathways involved, have not been studied in detail historically and thus remain unclear. 

However mounting evidence suggests that mechanical stimuli may be implicated in the promotion of stem cell-like characteristics, chemoresistance, EMT and metastasis [87]. Table 1 summarizes the research of mechanical stimuli and its influence on ovarian cancer progression and metastasis. 

## 7. Shear Stress in Ovarian Cancer Metastasis

Within the ovarian cancer patient population, approximately 40% of women develop ascites, a retainment of fluid within the peritoneal cavity [113]. The fluid buildup is caused by the blockage of typical lymphatic drainage mechanisms compounded by the increased permeability of surrounding vasculature allowing for the infiltration of cellular and protein components not normally present within this cavity [7,21,113]. The surplus fluid engulfs the ovaries and stimulates the primary tumor site with shear stresses from fluid movement caused by breathing, digestion, and general body movement [87] (Figure 1B,E). In ovarian cancer, ascites, tumor cells acquire elevated stem cell characteristics along with metastatic properties and enhanced chemoresistance in response to shear stress [88,89,90]. Ovarian cancer culture under continuous fluid flow has shown that shear stress acts to suppress E-cadherin expression via upregulation of vimentin, a hallmark of EMT [95,96,97,114] as well as increase the expression of p27^Kip1^, which is involved in proliferation and the cell cycle [115,116,117]. Studies under 2D stimulus continue to implicate shear stress with EMT, showing a prepotency for structural changes such as stress fiber formation and cell elongation in ovarian cancer [99]. Additionally, exposure to shear stress seems to promote chemoresistance in ovarian cancer [118]. For example, ATP-binding cassette transporters ABCG2/BCRP and P-glycoprotein, responsible for drug efflux from the cell [100,101], are upregulated when stimulated with shear stress, showing a dramatic reduction in the effectiveness of paclitaxel and carboplatin treatment [94]. Beyond increased drug resistance, cells also show enhanced cancer stem like cell populations expressing Oct-4, CD117 and CD44, which are tied to cancer dedifferentiation [119] metastasis [102] and self-renewal [88,103]. MicroRNA-199a-3p, which post-transcriptionally inhibits gene expression [120,121], was found to be upregulated and phosphatidylinositol 3-kinase/Akt pathways are also activated by shear [94]. Therefore, all the current evidence points towards the role of shear stress to increased EMT, which then leads to metastasis as cells detach and migrate to secondary sites for seeding nodules, attachment, and tumor re-growth.

## 8. Tensile Stress in Ovarian Cancer Metastasis

As a tumor grows and expands, it develops tensile forces, longitudinal stress on cells at the periphery of the developing mass (Figure 1C). The circumferential strain stems from the expanding interface of the tumor with the surrounding host environment [122,123]. As an example, it can be thought of as filling a balloon with air, as the interior expands the border cells are pressed outward and stretched via interaction with their surroundings. Tensile forces have been shown to activate the RhoA/ROCK signaling cascade and play a major role in VEGF-mediated angiogenesis in endothelial cells [104]. This pathway has been shown to be dysregulated in tumor-derived endothelial cells accounting for their inability to respond to uniaxial cyclic strain, reorient their cytoskeleton, and respond appropriately to ECM elasticity [124]. Physiological strain has also been found to influence cellular proliferation and drug treatment responses in lung cancer models cultured using the commercially available Flexcell system [106]. Though this form of stress is studied in a variety of solid tumors and other relevant organ systems no such ovarian specific study has been performed to the authors’ knowledge. The aforementioned lack of data might be due to the minimal amounts of tension predicted to be present in the ovarian TME when compared to respective compression and shear stress values. However, tensile forces are a worthy investigation to determine if critical pathways and phenotypic effects are maintained between cancer cell types and the influence of solid strain [125].

## 9. Compression Stress in Ovarian Cancer Metastasis

Compression is unique for ovarian cancer as it is caused by two primary sources (Figure 1D). The first of which is growth-induced stress, stemming from the aberrant cell proliferation displacing the native cell populations. The stress-induced from this form of growth was estimated to range between 4.7 and 18.9 kPa in human tumors and 0–1.3 kPa for avascular tumor spheroids via experimental data and mathematical models [122]. The studies by Jain et al. suggest that external stress, stemming from the native tissue, is a more noteworthy contributor to total perceived stress, as compared to the compressive stress from tumor growth that accounts for less than 30% of total solid stress of a tumor [122]. In addition to growth-induced stress, the presence of excess fluid and ascites creates hydrostatic pressure adding to the compressive forces experienced by ovarian cancer cells [92,126] though this contribution to compression is highly variable between patients and is dependent on the volume of ascitic fluid. Taken together it is evident that the influence of compression is no small part of the mechanobiology of ovarian cancer.

Investigations of the influence of compression on cancer progression have repeatedly found a tendency towards advanced invasive and metastatic phenotypes [127,128]. However, these studies are only beginning to investigate the ovarian cancer-specific response. Recently, Klymenko et al. modeled ascitic fluid pressure effects on ovarian cell line aggregates OVCA429, OVCA433, DOV13 and SKOV3 in both a pressurized chamber and Flexcell compression plus system [127]. Though they did not observe a change in proliferation, the compressive stimulus enhanced the invasion capability of the OVCA433 cells into hydrogels. Further, changes in EMT gene expression were found to depend on the duration of the compressive stimulus as 6 h down-regulated EMT genes and 24 h stimulation enhanced EMT gene expression in both DOV13 and OVCA433 cell line aggregates. Similarly, Burkhalter et al. found ovarian cancer compressive mechanotransduction acting through the WNT [108] pathway, which has been tied to carcinogenesis of a variety of ovarian cancer subtypes as well as, proliferation, chemoresistance, and stemness [129]. McGrail et al. found that cells utilize osmotic regulation to withstand and survive growth-induced solid stress in a variety of cancers including ovarian [109]. Beyond these studies, ovarian cancer is neglected in compression investigations compared to other cell types, such as breast and metastatic bone niches. While other forms of cancer including breast [128] and pancreatic [130], have shown enhanced invasion and metastatic tendencies due to solid stress [131], direct compressive forces have yet to be tied directly to enhanced cancer stem like cell prevalence. Further investigation is thus needed to determine if mechanotransduction findings are maintained within ovarian cancer when subjected to compression, as well as, if and how this prevalent mechanical stimulus influences the development of CSCs and metastasis.

## 10. Matrix Stiffness Stress in Ovarian Cancer Metastasis

As the cancer progresses, the innate ECM is also altered via the desmoplastic response [132], this leads to increased ECM stiffness within the tumor (Figure 1B). The buildup of scar-like tissue alters the stiffness and material properties of the microenvironment in which the cells reside, contributing to the variety of mechanical stimuli that activate a diverse set of mechanotransduction pathways. ECM stiffness has a direct impact on the formation and response of the cellular cytoskeleton. The cell counterbalances external resistive forces within the cytoskeleton leading to numerous downstream effects. When considering the dimensionality and stiffness of the experimental culture, an additional set of phenotypes and activation pathways come to light. Preference to softer substrates for proliferation, EMT, elongation, and chemoresistance have been demonstrated on 2D surfaces [109] for ovarian cancer which has implications in metastasis and clinical outcomes. The expression of these observed phenotypes depended entirely on the Rho/Rock pathway and its enhancement or inhibition [109]. Within a 3D environment, ovarian cancer morphology tends toward spheroid formation and has enhanced chemoresistance and once again prefers compliant substrates for proliferation [92]. The stiffness of the ovarian cancer cell itself may be a useful biomarker when screening for CSCs as ovarian cancer stem-like cells show an increased ability to undergo deformation of up to 72% when compared to ovarian cancer controls [133]. While not yet shown in ovarian cancer, stiffness in other cancers such as colorectal cancer [134], breast cancer, gastric cancer, and osteosarcoma [135] can affect CSC phenotypes. These data suggest that stiffness could also influence metastasis in ovarian cancer by promoting more metastatic CSC phenotypes, though this still needs to be investigated. Therefore, the literature implies that there might be a direct effect of increased stiffness on the displacement of native ovarian tissue cells, and tumor growth and metastasis.

## 11. Stress Relaxation, Creep Compliance in Ovarian Cancer Metastasis

Biological and synthetic ECMs are often viscoelastic materials, whereby the component’s stiffness is dependent on both stress and stress duration. A material’s creep compliance is a metric measuring the rate at which strain increases for an applied and constant stress. Meanwhile, a material’s stress relaxation is a metric relating the rate of stress reduction to a given and constant strain. Depending on whether cells are exposed to viscoelastic or elastic matrices will determine cellular behaviors such as spreading, proliferation, and remodeling [108]. The viscoelastic matrices are characterized by their ability to dissipate forces exerted by embedded cells through matrix yielding and remodeling [108] (Figure 2). Comparatively, fully elastic matrices store the forces exerted by the cell and no remodeling occurs, and do not depend on loading rates [136]. Recent studies have shown 3T3 mouse fibroblasts show an increased ability to plastically reorganize and realign collagen fiber networks in response to stress relaxation and that faster matrix stress relaxation times, results in increased cell spreading and proliferation of 3T3 fibroblasts and mesenchymal stem cells [110,136].

There are currently no published studies investigating the effects of viscoelasticity on ovarian cancer at the time of this publication. However, recent studies have shown that viscoelastic mechanistic responses are present within various cancer and cancer-associated cells. For example, breast cancer cells cultured in viscoelastic high plasticity hydrogels show an increased propensity to migrate using actin mediated force to extend and contract invadopodia compared to cells cultured in low plasticity hydrogels. The cells migrate using the invadopodia to plastically alter the surrounding matrix over cells cultured in low plasticity hydrogels and open wider pore in the matrix to move through [111]. Additionally, changes in intracellular viscoelasticity have been associated with cell malignancy, and that metastatic cells are softer compared to benign counterparts [137,138]. A study by Wullkopf et al., showed that breast cancer and pancreatic cancer cells at the tip of spheroids invading 3D matrices have higher viscosities than cells located at the center of the spheroids [112]. The increase in viscosity from the center to the protrusion tip supports the hypothesis that more compliant cells have higher motility through confined environments [130]. Furthermore, another study has shown malignant breast tumors can be differentiated from benign breast tumors by a difference in their viscoelastic properties using magnetic resonance elastography, suggesting a relationship between viscoelasticity and cancer progression [139]. There is evidence to suggest that careful consideration of tumor microenvironmental extracellular matrix viscoelastic properties, as well as, the viscoelastic properties of the cells themselves may be useful when devising in vitro studies, new drug treatments, and early detection methods.

## 12. Challenges Confronting Future Metastasis Research in Ovarian Cancers

Thus far we have reviewed mechanisms of ovarian cancer metastasis and discussed the involvement of both cancer stem-like cells and mechanical stimuli in this process. While there are clear connections between CSCs, mechanical stimuli, and metastasis, a lot is still unknown about how they are intertwined, as a result of significant research challenges. In this last section of the review, we discuss the main hurdles facing CSC and mechanotransduction research in ovarian cancer metastasis studies and offer suggestions to overcome these roadblocks.

## 13. Challenges with Incorporating CSCs in Metastasis Research

CSCs are associated with increased metastatic capacity and would be most able to initiate secondary tumors due to their tumorigenicity. The role of CSCs in ovarian cancer metastasis is emphasized by the ascites fluid, which contains free-floating spheroids enriched in CSCs. The spheroids are significant as CSCs are able to resist anoikis in non-adherent environments. The increased migratory capacity, chemoresistance, and ability to colonize that characterize CSCs are critical to metastasis and thus poor clinical outcomes. However, the lack of clearly defined ovarian CSC populations and the unknown role of the various putative CSC subpopulations in metastasis relative to one another remains a challenge in CSC focused metastasis research. A clearer understanding of ovarian CSCs in metastasis is essential to the development of new CSC targeting therapies to prevent and treat metastasis. Given that chemotherapies such as carboplatin and cisplatin can promote CSC populations and metastasis, improved understanding of CSCs and metastasis also has implications in primary treatment options to reduce the incidence of relapse.

In order to incorporate CSCs in basic biology and translational research focused on metastasis, the first step is solidifying the heterogeneity in CSC populations by examining the expression of all or most CSC markers in all experiments. This will help to explain discrepancies found between different labs and experimental models of metastasis. It may also help to elucidate the role of the various CSC subpopulations at different stages of metastasis and tumor progression. Along this vein, we recommend the use of more than one CSC marker to isolate CSC populations to improve robustness of the cancer stem like label. We additionally propose the use of functional assays such as sphere-forming ability, differentiation capacity, and chemoresistance assays to ensure that isolated cells are in fact more stem like than their counterparts. Furthermore, the plasticity inherent in CSCs provides an additional challenge in CSC based metastasis research as their marker expression could change rapidly making them difficult to isolate and target. However, with a more comprehensive knowledge of CSC differentiation patterns and CSC marker hierarchies obtained through a comprehensive examination of different CSC marker subsets and their functionalities, the changes in CSC marker expression could conceivably be predicted and used as a basis for therapy.

Additionally, heterogeneity in ovarian cancer exists in not only its histological subtypes but also in the intratumoral heterogeneity of CSCs. This poses a big problem in identifying markers, which are examined from a small piece of the tumor thus adding to the challenges of researching the role of CSCs in metastasis and clinical outcomes. More thorough clinical evaluation of CSC marker expression using single-cell sequencing techniques to get a better idea of heterogeneity of CSC populations within a whole tumor and between patients will help us to better predict their relationship to metastasis and clinical outcomes to ultimately develop better therapeutics.

Finally, the inclusion of fully characterized heterogeneous patient samples within in vitro models will help to elucidate the role of different heterogeneous cell compositions in promoting cancer stem-like cells and metastasis. This is especially important to consider in studies involving both CSCs and mechanical stimuli in metastasis as the cell composition can affect the mechanical environment through TME remodeling and extracellular matrix protein secretion [61], thereby influencing CSCs and thus metastatic potential through soluble and mechanical stimuli. More details regarding the challenges of mechanical stimuli research in ovarian cancer and suggestions for improvement are discussed in the next section.

## 14. Challenges with Physiologic In Vitro and In Vivo Models to Study Mechanical Stimuli in Ovarian Cancers

Ovarian cancer pathology is not completely understood, hindering the development of improved treatments and early detection techniques that may identify the disease before metastatic spread. Therefore, to improve clinical outcomes a better understanding of the disease is imperative. A lot needs to be learned about the role of mechanotransduction in ovarian cancer, as there are minimal experiments available to draw conclusions from. Particularly, extensive research on how stiffness affects ovarian cancer needs to be carried out in detail. While results showing promotion of EMT, chemoresistance and cancer stem cell surface markers is evident in few studies that have been carried out to model shear stress in ovarian cancer, more studies are needed. The main challenges facing the advancement of ovarian cancer mechanotransduction research in relation to metastasis are: (1) the inability to detect the disease early and thus study its progression and (2) the lack of physiological measures of mechanical forces tumor cells experience across a large sample population.

The lack of early detection mechanisms limits treatment options as well as, researchers’ understanding of the progression and changing mechanical influences of the disease. How initial forces aid in the advancement of the disease have only been hypothesized and studied in vitro where external forces are largely estimated from computer models and other solid tumor measurements. To explore the effects of mechanotransduction on ovarian cancer in detail, there is an urgent need for the development of physiologically relevant in vitro 3D models that can recapitulate the mechanical forces seen in ovarian cancer in vivo. For example, continued development of 3D in vitro mechanical stress bioreactors that allow for isolation of different stresses will facilitate the study of their individual effects on disease progression, metastasis, and drug response. Moreover, as 3D in vitro models become more advanced, it will be important to incorporate heterogeneous cell populations within the bioreactors as different cell compositions can influence the mechanical environment through ECM protein secretion and TME remodeling thereby altering the metastatic capability of the cells being studied.

Furthermore, ovarian cancer resides in a unique microenvironment that is believed to change dynamically with the progression of the disease. Currently, the mechanical force estimates that cells are predicted to experience have not been measured in vivo which limits our understanding of the mechanical microenvironment, as well as the accuracy of in vitro models. With precise measurements of pathophysiologic compressive forces, solid strain, ECM stiffness, and shear stress throughout disease progression, we would obtain a more complete picture of the most relevant influences on ovarian cancer cells. Additionally, widespread sampling should be conducted to complete demographics across stages of the disease. Armed with this knowledge researchers would be more equipped to isolate and target specific mechanotransduction pathways distinct to a patient’s condition and most likely to aid in her treatment response.

## 15. Conclusions

Metastasis is the deadliest feature of ovarian cancer, yet there is currently no therapy to combat this process as a result of our incomplete understanding of the complex interplay between its various driving factors. Ovarian cancer stem-like cells and the mechanical stimuli within the tumor microenvironment have emerged as two key features in driving metastasis suggesting that they could serve as suitable prognostic indicators and targets for therapies preventing or combating metastasis. Despite significant evidence that CSCs and mechanical stimuli are involved in ovarian cancer metastasis, and may even be intertwined through the mechanical promotion of CSC phenotypes, much is yet unknown about their roles in metastasis.

This gap in knowledge is due in part to the lack of methods to measure physiological mechanical forces over time as metastasis progresses. Furthermore, the plasticity of CSCs and the heterogeneity of CSC markers within tumors and between patients has hindered discovery of the roles of different CSC subpopulations in the progression of ovarian cancer metastasis. To address these challenges, we propose: (1) more robust characterization of CSC heterogeneity within tumors and between patients through evaluation of a comprehensive panel of putative CSC markers in all CSC experiments and clinical evaluations; (2) the development of methodologies for early detection of ovarian cancer to facilitate the measurement of mechanical forces and changes in CSC populations over time; (3) the development of techniques to measure physiological mechanical forces. In summary, the studies reviewed here indicate that CSCs and mechanical stimuli, are crucial to the development of a tumorigenic, platinum resistant, metastatic phenotype of ovarian cancer, and therefore warrant further mechanistic studies.

## Figures and Tables

**Figure 1 cancers-11-01008-f001:**
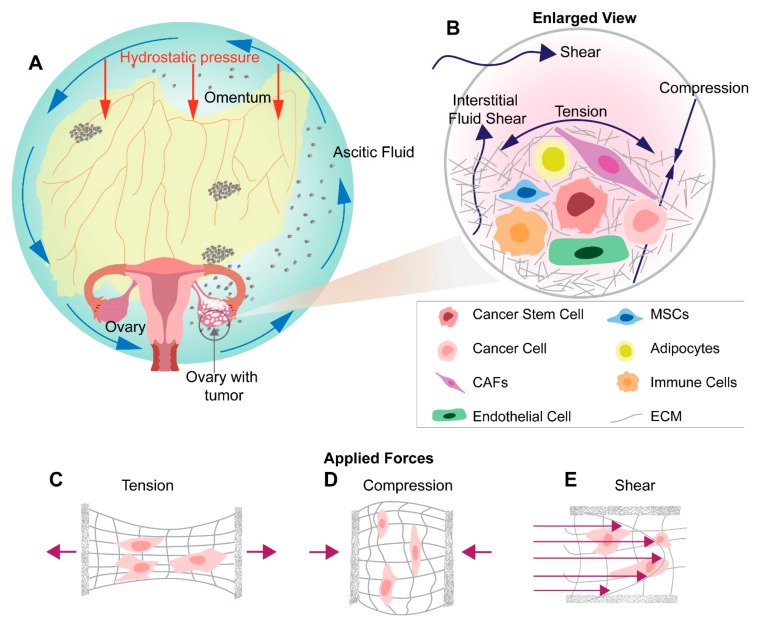
Mechanical and cellular regulators of ovarian cancer metastasis. Ovarian cancer progression and metastasis are driven by chronic exposure to mechanical stimuli and tumor supporting cells. (**A**) Increased fluid movement within the pathological ovarian tumor microenvironment exposes ovarian cancer cells to high levels of shear stress. The aforementioned environment is caused by a combination of normal body function and ascitic buildup within the peritoneal cavity. Teal circle indicates the peritoneal cavity, blue arrows indicate interstitial fluid flow and red arrows indicate hydrostatic pressure. (**B**) Zoomed in view of the cellular makeup of the tumor microenvironment, which is composed of cancer stem-like cells, cancer cells, cancer-associated fibroblasts (CAFs), endothelial cells, mesenchymal cells (MSCs), adipocytes and immune cells, that interact to influence tumor progression and metastasis. Neoplastic growth steadily increases tumor size providing radial tension and axial compression to ovarian cancer cells. Effects of (**C**) tension, (**D**) compression and (**E**) shear stresses on cancer cells within the ovarian tumor microenvironment.

**Figure 2 cancers-11-01008-f002:**
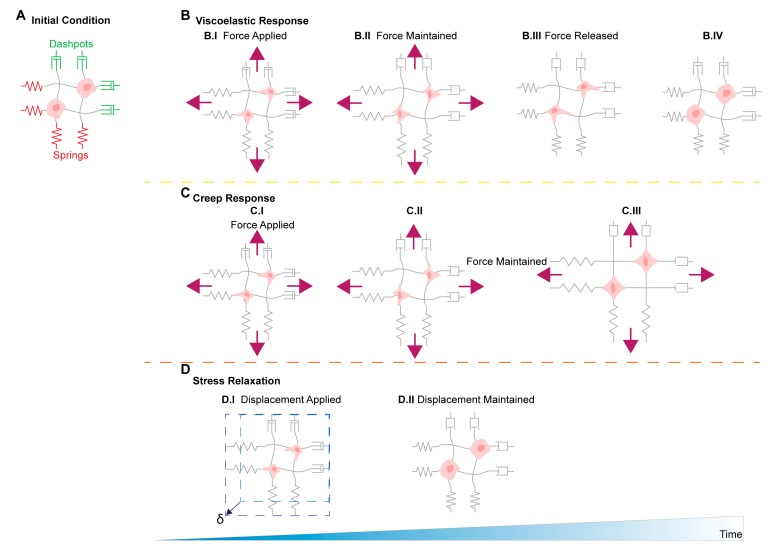
Spring and dashpot model to represent the viscoelastic response, creep response, and stress relaxation of the extracellular matrix in the tumor microenvironment. (**A**) Initial unstressed cellular ECM is represented by spring (elastic) and dashpot (viscous, time-dependent) model response to applied force and deformations. (**B**) Viscoelastic response characteristics of ECM. (**B.I**) Initial force application creates immediate elastic stretching of matrix components as shown through elongation of the spring components. (**B.II**) Maintenance of applied force continues to deform components in a time-dependent manner as represented by dashpots. (**B.III**) Release of force application allows for immediate elastic recoil in some components as indicated by spring recoil and time-dependent restructuring indicated by prolonged dashpot elongation. (**B.IV**) Prolonged release of force allows for complete recovery of elastic deformation and realignment of ECM structures. (**C**) Creep response. (**C.I**) Initial force application creates immediate elastic stretching of matrix components as shown through elongation of the spring components. (**C.II**) The continued application of force for prolonged time continues to deform ECM components as seen in the elongated dashpots. (**C.III**) The prolonged force stimulus denatures the ECM components creating plastic deformation or creep in the polymer networks that would not be recovered with the removal of the force. (**D**) Stress relaxation. (**D.I**) A set displacement is applied on the ECM components and no additional forces are maintained. An immediate response from elastic spring-like components stretches the material. (**D.II**) As deformation is held constant, internal stress are reduced over time as the ECM relaxes. This is mechanically demonstrated as elongation of the dashpot components, which allows for recoil in the elastic-like materials shown here as spring conformational change into resting state.

**Table 1 cancers-11-01008-t001:** Summary of studies on mechanical stimulation in ovarian cancers and consequences on metastasis and cancer progression.

Mechanical Stimuli	Physiological Sources of Mechanical Stimuli in Ovarian Cancer	Influences of Mechanical Stimuli on Cancer Progression and Metastasis
**Shear**	Ascitic buildup and fluid movement Interstitial fluid flow Vascular/arterial blood flow	Suppression of E-cadherin and upregulation of vimentin [95,96,97] Increased p27-kip1 [98] Stress fiber formation and cell elongation [99] Chemoresistance–ABCG2 and P-gp upregulation [94,100,101] Enhanced cancer stem cell populations (Oct4, CD177, CD44 [88,102,103] MicroRNA-199a-3p upregulation and Phosphatidylinositol3-kinase/Akt pathway activation [94]
**Tension/Strain**	Tumor expansion against ECM and native cell populations Circumferential strain	**Non-Ovarian Cancer Findings** Activation of Rho/ROCK pathway and VEGF-mediated angiogenesis [104,105] Proliferation and chemoresistance [106]
**Compression**	Tumor expansion and displacement of surrounding ECM Hydrostatic pressure from ascites	Enhanced invasion [107] Alteration in EMT gene expression [107] WNT pathway alterations [108]
**Stiffness**	Desmoplastic response Elevated ECM synthesis and remodeling	Enhanced proliferation, EMT expression, cellular elongation, chemoresistance on soft substrates [109] Rho/Rock pathway dependence [109]
**Others: Stress relaxation, Creep compliance**	**ECM restructuring from:** Degradation by cells Remodeling and synthesis of additional ECM elements Desmoplastic response	**Non-Ovarian Cancer Findings** Plastic reorganization and realignment of ECM [110] Migration (actin mediated) via invadopodia [111] Alteration in invasive potential [112]

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
