# Peer review of "The Role of Cancer Stem Cells and Mechanical Forces in Ovarian Cancer Metastasis"

_cancers, 2019, doi:10.3390/cancers11071008_

Round 1

Reviewer 1 Report

Mehta et al., provide a comprehensive and well-written review of ovarian cancer stem cells and the role of mechanical stress on stem cells and ovarian cancer metastasis. This review will be helpful for the scientific community

Two minor concerns:

1) In the sections on mechanical stress, the authors are careful to explain which data come from ovarian cancer studies and which data come from other types of cancer.  They are not so transparent in the section on ovarian cancer stem cells.  For example, on lines 243-248 they discuss the importance of CAFs on TME and they cite Murata, et al., (citation 54).  This study was done using cervical cancer cells and may not be applicable to ovarian cancer. Authors should make it clear (as they do in the mechanical stress section) when the results they are reviewing were done using other cancer types, not ovarian cancer.  Another example is in the paragraph on lines 280 – 295. They cite Karnezis (citation 70) concerning lymphatic dissemination and Hoshino (citation 75) concerning exosomes role in metastases. Neither of these studies were using ovarian cancer samples. They end this paragraph with the sentence, "Thus, all the examples described in this section exhibit the various CSC niche dependent bidirectional interactions between ovarianCSCs and the TME elements."  Which is not entirely true as several of these examples were using CSCs from non-ovarian models.  I did not check all the references, but, in general, authors should state whether or not the findings being cited are ovarian specific, as there are important differences in stem cell behavior between different organs and cancer.

2) Adipocytes is misspelled in the diagram as Adopocytes.

Author Response

The authors would like to thank the editors and the reviewers for their time that they expended to review our manuscript, and appreciate their insightful and helpful comments. We have made revisions to the manuscript based on your feedback. These changes are highlighted in red font colorin the marked manuscript. A detailed list of edits and the author response to the reviewers’ comments follow below in red font color

Comments from the editors and reviewers: 

Reviewer #1:

Two minor concerns:

1)            In the sections on mechanical stress, the authors are careful to explain which data come from ovarian cancer studies and which data come from other types of cancer.  They are not so transparent in the section on ovarian cancer stem cells.  For example, on lines 243-248 they discuss the importance of CAFs on TME and they cite Murata, et al., (citation 54).  This study was done using cervical cancer cells and may not be applicable to ovarian cancer. Authors should make it clear (as they do in the mechanical stress section) when the results they are reviewing were done using other cancer types, not ovarian cancer.  Another example is in the paragraph on lines 280 – 295. They cite Karnezis (citation 70) concerning lymphatic dissemination and Hoshino (citation 75) concerning exosomes role in metastases. Neither of these studies were using ovarian cancer samples. They end this paragraph with the sentence, "Thus, all the examples described in this section exhibit the various CSC niche dependent bidirectional interactions between ovarian CSCs and the TME elements."  Which is not entirely true as several of these examples were using CSCs from non-ovarian models.  I did not check all the references, but, in general, authors should state whether or not the findings being cited are ovarian specific, as there are important differences in stem cell behavior between different organs and cancer.

Thank you for your astute comment! We have edited this section, included mentions of other cancers where applicable, and made the article more ovarian cancer specific. These changes are reflected in page 6 222-236, page 7 243-250, and page 8 285-287, 289-290, 294-295, 298-300. 

2)            Adipocytes is misspelled in the diagram as Adopocytes.

 Thank you for the keen observation! We have corrected the spelling. 

Reviewer 2 Report

The authors provide robust description of how ovarian cancer dissemination differs from other types of cancers, citing many of the relevant studies. The review is well written and informative in this respect citing much of the relevant literature. With the focus on mechanical stimulation, I think the review should address a few other relevant points in this area. Also, other necessary corrections are listed below. Figure 1 also seems somewhat inaccurate in terms of the force profiles, which are important in discussing the relevant mechanical stimuli.

Figure 1 schematics on force profiles and response to external stress are very confusing. The directions of the arrows in Figure 1 D-K seem somewhat random and not always factual based on the images. For example stress in F would vary with distance from the wall and be in the opposite direction of flow. In J, it looks like stress relaxation results in tension, not clear why that would be the case.  Also, how elastic vs. viscoelastic are represented in I-J is very unclear. If the deformations are responses to external stress then the external stress should be shown. Also, elastic substrates do deform in response to stress but the deformation is reversible, and your image shows elastic matrix response is not deformed with stresses that don’t appear to be balanced.

Page 6, lines 228-231, unclear what you mean by erratic organ, this part should be clarified.

Also, some of the references are incomplete, including number 53 (no journal information) and 55 (no article title). All references should be checked to ensure they comply with journal guidelines for bibliography. Note reference 53 refers to paper on “CLIC4 mediates TGF-beta1-induced…” and the text refers to TGFs1 incorrectly (line 239-40, page 6).

P7, lines 274-79 discuss anti-tumor immunity referring to MDSC, macrophage, and Treg being anti-tumor. MDSC and Treg are both involved in cancer immunosuppression, and macrophages have pro- and anti- tumor roles depending on their polarization.

Format for Table 1 needs some improvement. Some references are described very generally and other more specifically, I think this should be more consistent. Also, under shear some of these articles are on breast cancer but that is not highlighted like in the other sections. Bulleting and spacing are also inconsistent making the table look kind of unprofessional.

Papers referenced for P27Kip1 seem kind of odd, not sure if this is the focus of these papers, maybe want to check references (also, it seems like more seminal papers on this could be cited). Also CDKN1B plays role in cellular senescence and growth arrest in p1, is this limiting the growth of CSCs making them more quiescent?

Page 11 refers to high compression due to hydrostatic pressure and excess fluid, which seems reasonable, but this also varies from patient to patient and depending on whether patients have had paracentesis, so maybe this should be mentioned. \

P11 second paragraph talks about influences on compression and cancer progression. Some studies on mechanical compression of ovarian cancer spheroids are mentioned, but osmotic regulation is not mentioned and Biophys J paper from McGrail not cited.

P11 bottom paragraph, starting with line 428. The implications of acute and chronic stress are not addressed. Very little discussion on stress relaxation and creep compliance, which are keywords listed for this paper, so should be somewhat of a focus of this paper. I would suggest expanding this discussion or changing the focus. If you want to talk about acute and chronic/long term stress, what are contributers to these stresses in ovarian cancer?

Not clear that EpCAM is a cancer stem cell marker.

P13 challenges should include heterogeneity in ovarian cancer cells and their stroma which makes it difficult to predict cell response to mechanical stress. Models not only need to account for complex mechanics but also heterogeneity in cells and their surrounding environments.

Author Response

The authors would like to thank the editors and the reviewers for their time that they expended to review our manuscript, and appreciate their insightful and helpful comments. We have made revisions to the manuscript based on your feedback. These changes are highlighted in red font colorin the marked manuscript. A detailed list of edits and the author response to the reviewers’ comments follow below in red font color

Reviewer 2: 

The authors provide robust description of how ovarian cancer dissemination differs from other types of cancers, citing many of the relevant studies. The review is well written and informative in this respect citing much of the relevant literature. With the focus on mechanical stimulation, I think the review should address a few other relevant points in this area. Also, other necessary corrections are listed below. Figure 1 also seems somewhat inaccurate in terms of the force profiles, which are important in discussing the relevant mechanical stimuli.

Figure 1 schematics on force profiles and response to external stress are very confusing. The directions of the arrows in Figure 1 D-K seem somewhat random and not always factual based on the images. For example, stress in F would vary with distance from the wall and be in the opposite direction of flow. In J, it looks like stress relaxation results in tension, not clear why that would be the case.  Also, how elastic vs. viscoelastic are represented in I-J is very unclear. If the deformations are responses to external stress then the external stress should be shown. Also, elastic substrates do deform in response to stress but the deformation is reversible, and your image shows elastic matrix response is not deformed with stresses that don’t appear to be balanced.

Thank you for your insightful comments! We have changed Figure 1 based on the reviewer’s comments. We now also include Figure 2 to accurately portray the viscoelastic properties of the extracellular matrix. In Figure 2, we use the spring (elastic) and dashpot (viscous, time dependent response) model to convey the effect of viscoelastic forces on cells. 

Page 6, lines 228-231, unclear what you mean by erratic organ, this part should be clarified.

We have clarified this statement in the revised text, page 6, line 219-220.

Also, some of the references are incomplete, including number 53 (no journal information) and 55 (no article title). All references should be checked to ensure they comply with journal guidelines for bibliography. Note reference 53 refers to paper on “CLIC4 mediates TGF-beta1-induced…” and the text refers to TGFs1 incorrectly (line 239-40, page 6).

Thank you for the astute observation! We have edited this text.

We experienced some errors during compiling references using multiple bibliography software. We have now re-inserted all of the references, and followed the journal guidelines for bibliography. 

P7, lines 274-79 discuss anti-tumor immunity referring to MDSC, macrophage, and Treg being anti-tumor. MDSC and Treg are both involved in cancer immunosuppression, and macrophages have pro- and anti- tumor roles depending on their polarization. 

Thank you for this comment! We believe that it is not the scope of our manuscript to discuss the complex relationships involved in tumor immunomodulation, and the pro- and anti-tumoral roles of various immune cells. We have modified this sentence by making it clearer that immunosuppression has an important role in creating a pre-metastatic niche for tumor cells (page 8, line 269- 281). 

Format for Table 1 needs some improvement. Some references are described very generally and other more specifically, I think this should be more consistent. Also, under shear some of these articles are on breast cancer but that is not highlighted like in the other sections. Bulleting and spacing are also inconsistent making the table look kind of unprofessional. 

We have formatted and improved the Table in response to the reviewer’s comment.

Papers referenced for P27Kip1 seem kind of odd, not sure if this is the focus of these papers, maybe want to check references (also, it seems like more seminal papers on this could be cited). Also, CDKN1B plays role in cellular senescence and growth arrest in p1, is this limiting the growth of CSCs making them more quiescent?

As requested, we have cited more seminal and highly referenced papers to showcase the implications of P27Kip1 on cell cycle progression and cellular proliferation. The following papers have now been referenced: 

1. Kossatz, U. et al. Skp2-dependent degradation of p27kip1 is essential for cell cycle progression. Genes Dev. 18, 2602–2607 (2004).

2. Doetsch, F. et al. Lack of the Cell-Cycle Inhibitor p27Kip1 Results in Selective Increase of Transit-Amplifying Cells for Adult Neurogenesis. J. Neurosci. 22, 2255–2264 (2002).

3. Chen, D. et al. Downregulation of cyclin-dependent kinase 2 activity and cyclin A promoter activity in vascular smooth muscle cells by p27(KIP1), an inhibitor of neointima formation in the rat carotid artery. J Clin Invest 99, 2334–2341 (1997).

We also appreciate the reviewer’s suggestion concerning CDKN1B, however this is still an open question. The valuable insight for future investigation is appreciated and we hope to generate similar responses and novel ideas from the readers of this article. 

Page 11 refers to high compression due to hydrostatic pressure and excess fluid, which seems reasonable, but this also varies from patient to patient and depending on whether patients have had paracentesis, so maybe this should be mentioned. 

We have addressed this comment on page 12, line 409-410.

P11 second paragraph talks about influences on compression and cancer progression. Some studies on mechanical compression of ovarian cancer spheroids are mentioned, but osmotic regulation is not mentioned and Biophys J paper from McGrail not cited.

We now mention the McGrail research study, and cite their paper. 

P11 bottom paragraph, starting with line 428. The implications of acute and chronic stress are not addressed. Very little discussion on stress relaxation and creep compliance, which are keywords listed for this paper, so should be somewhat of a focus of this paper. I would suggest expanding this discussion or changing the focus. If you want to talk about acute and chronic/long term stress, what are contributers to these stresses in ovarian cancer?

Per reviewer comments, we have removed stress relaxation and creep compliance from the keyword list. As the investigation of how both stress relaxation and creep compliance affect cancer cells is still not very well established, we aim to alert the readers that these variables are in fact present within the extracellular matrix. We further aim to highlight seminal work showing that cells are impacted by these material properties and that extracellular matrix/substrate viscoelasticity is physiologically relevant, and therefore an important consideration for cancer researchers studying extracellular matrix. We have added two definitions on lines 479-481, with material science in mind, to improve reader understanding of stress relaxation and creep compliance.

We would define chronic exposure to a stimulation as lasting more than 3 months. This could be a symptom of a disease, an adverse environment, or a mechanical stress with continued and/or recurring contact. One overarching theme of this paper is that chronic exposure to elevated mechanical stimulation is a key driving force in cancer metastasis and stemness traits in vivo. We aim to show that certain mechanical stimuli in the tumor microenvironment become elevated over time, reaching a state that is both aberrant and chronic in nature. To address this issue, we have added a clarification on page 13, lines 449-452. 

Viscoelasticity is one of the mechanical forces that we believe merits further investigation. It should be noted that most in vitrostudies are done on the time frame of 72 hours (often closer to 12) and yet scientists still see the impacts of a heightened mechanical microenvironment on cancer progression. In this regard, experiments performed on cells in vitromight be considered acute stress exposure. We believe that the startling results from the field of cancer mechanobiology may help to explain the progression of ovarian cancers as seen by oncologists worldwide, and that chronic exposure to aberrant stresses may be the smoking gun for metastasis. We hope to have adequately addressed the reviewer’s concern with the above edits to the manuscript.

Not clear that EpCAM is a cancer stem cell marker.

Thank you for pointing that out!  We have removed EpCAM reference from this section.  

P13 challenges should include heterogeneity in ovarian cancer cells and their stroma which makes it difficult to predict cell response to mechanical stress. Models not only need to account for complex mechanics but also heterogeneity in cells and their surrounding environments.

Thank you for bringing up this excellent point! We have added a paragraph addressing this at the end of the ‘Challenges with incorporating CSCs in metastasis research’ section since the heterogeneous cell populations changing the mechanical environment may also influence CSC populations. A section discussing this in the ‘Challenges with physiologic in vitroand in vivomodels to study mechanical stimuli in ovarian cancers’, has been added to this part of the review, in page 16, lines 579-582.

Reviewer 3 Report

Paper outline:

In the manuscript entitled ’The Role of Mechanical Stimulation and Cancer Stem Cells on Ovarian Cancer Metastasis’ by  Michael E. Bregenzer , Eric Horst , Pooja Mehta , Caymen Novak , Taylor Repetto , Geeta Mehta , the current literature on mechanics and cancer stem cells in ovarian cancer metastasis is reviewed. First ovarian cancer is introduced, after which knowledge on the process of metastasis in ovarian cancer is explained. The roles of CSCs in supporting metastasis are highlighted, followed by discussion of mechanical stimuli for ovarian cancer metastasis. The paper ends with a final statement in the form of an overall conclusion, in which 1) CSCs and mechanics are recognized as two key parameters involved in ovarian cancer metastasis, 2) a restrictive lack of knowledge is highlighted, and 3) recommendations are made for future research.

The review would fit well within the scope of the journal and in my opinion could be of interest to the journal’s readers. The writing is clear and concise and the spelling and grammar meet the required standard, although at various points in the manuscript the authors should have been more meticulous (see minor comments). Even though the topics of mechanics and stem cells have been reviewed previously in the context of ovarian cancer, the connection between the two in the context of metastasis could give a fresh perspective. To achieve this, I think the authors should consider the following:

Major comments:

1.       A vast issue with respect to cancer stem cells are the inconsistencies in the field. Nobody has actually isolated a well-defined cell population reproducibly. This makes it impossible to attribute properties, such as chemoresistance and invasiveness to one specific population, regardless of tumour type. The current manuscript also highlights this as the paragraph on ‘The roles of ovarian cancer stem like cells (CSCs) in metastasis’ exemplifies the inability to attribute a single set of markers to CSCs. Combined with the known cellular plasticity of marker expression, the overall view is that the concept of cancer stem cells may need to abandoned all together. To me, it does not suffice calling them cancer stem like cells in the manuscript. It still confers properties onto cells they may not have and vice versa. Overall, I think the authors should at least mention these issues with respect to CSCs in their review and give a critical view on the situation.

2.       The coherence in the manuscript between the section on CSCs and mechanics is currently achieved mainly through one review paper, in which I was not able to find influence of CSC on mechanics and vice versa. I think the manuscript would be improved is this connection between the two topics could be strengthened, as now it reads almost as two separate manuscripts: one on CSCs and one on mechanics.

3.       Figure 1 contains some elements that deserve improvement. In panel A, it is not completely clear to me what the big circle represents. Is this the peritoneal cavity, with the arrows representing fluid flow or pressure and the small blobs representing cancer cells? I would also suggest changing Panel B, as in its current format, the figure implies cells are randomly dispersed in ECM. If the only goal with Panel B is to show the different cell types present, the legend suffices. Panel C seems to be an alternative representation of Panel D-F and could therefore be omitted. Otherwise, clarification is needed on what the streaks in the circle in C represent. In Panels D-F, detail on the causes of the mechanics would be helpful (i.e. what do the grey bars represent?). The legend of Figure 1F (line 151) reads: ‘… cause genotypic and phenotypic changes’. This is not represented in the figure and should therefore not be mentioned.

4.       Accuracy of referencing should be enhanced. For example, in the introductory paragraph, ovarian cancer statistics are discussed, but the references accompanying these numbers are not actual statistics papers. Overall, the authors reference other review papers in various instances, making it difficult for the reader to retrieve the actual scientific data that support the statements made.  Furthermore, in line 250, the authors refer to ref 55 ‘as we described previously’, I could not find any authors on that paper matching with the current authors. Please explain use of the word ‘we’.

5.       The use of headings and sub-headings could be improved. 1) It is currently not clear where one section ends and the other begins: e.g. is ‘Interactions between ovarian cancer stem like cells and ovarian tumor microenvironment that modulate metastasis’ part of ‘The roles of ovarian cancer stem like cells (CSC) in metastasis’ or a separate paragraph? Is ‘Shear stress in ovarian cancer metastasis’ part of ‘Mechanical stimuli in ovarian tumor microenvironment and metastasis’ or not? Use of sub-headings would be appropriate if they sub-paragraphs. 2) Some of the headings are misleading in terms of content: e.g. the paragraph entitled ‘Tensile stress in ovarian cancers metastasis’ contains no information on ovarian cancer as ‘no such ovarian specific study has been performed’ and the paragraph on ‘Stress relaxation, creep compliance in ovarian cancers metastasis’  also continues into ‘There are currently no published studies investigating the effects of viscoelasticity on ovarian cancer at the time of this publication’.

6.       The last paragraphs ‘In vivo studies on ovarian cancer metastasis’, ‘Challenges with incorporating CSCs in metastasis research’ and  ‘Challenges with better in vitro and in vivo models to study mechanical stimuli’ are, in my opinion, not that well integrated in the manuscript. Now, these paragraphs come across as a sort of encore.

Minor comments:

1.       Figure 1 B, the cell legend on the right: CAFs is spelled as CAF’s (see also my next comment) and Adipocytes is misspelled.

2.       Line 232-237: Three different variants of spelling CAFs (CAFs, CAF’s and CAFS) are used.

3.       Line 224: Rephrase the sentence starting with ‘To address…’, it is not flowing well.

4.       Line 244: ‘Elucidated’ seems an odd choice of words in this context. Suggest replacing with ‘Illustrated’.

5.       Line 277: ‘…as well as hypoxia,’. It is not clear to me in what context hypoxia is mentioned here.

6.       Line 340-343, this sentence needs referencing.

7.       In line 349 ABCG2 is used, in line 463 ABCG2/BCRP, both are correct, but consistency in terms used would be good.

8.       Line 409 and 428: ‘cancers’ should read ‘cancer’.

9.       The formatting of the references is inconsistent, the majority of references are therefore lacking details (such as journal, volume, pages, some refs even lack a title).

10.   I would advise to reorganize the title. If CSCs are discussed before mechanics, I would also expect this to be reflected in the title. I would also replace ‘mechanical stimulation’ with ‘mechanics’ as the former may imply the discussion of how cells can be experimentally influenced.

Author Response

The authors would like to thank the editors and the reviewers for their time that they expended to review our manuscript, and appreciate their insightful and helpful comments. We have made revisions to the manuscript based on your feedback. These changes are highlighted in red font colorin the marked manuscript. A detailed list of edits and the author response to the reviewers’ comments follow below in red font color

Reviewer 3: 

Paper outline:

In the manuscript entitled ’The Role of Mechanical Stimulation and Cancer Stem Cells on Ovarian Cancer Metastasis’ by  Michael E. Bregenzer , Eric Horst , Pooja Mehta , Caymen Novak , Taylor Repetto , Geeta Mehta , the current literature on mechanics and cancer stem cells in ovarian cancer metastasis is reviewed. First ovarian cancer is introduced, after which knowledge on the process of metastasis in ovarian cancer is explained. The roles of CSCs in supporting metastasis are highlighted, followed by discussion of mechanical stimuli for ovarian cancer metastasis. The paper ends with a final statement in the form of an overall conclusion, in which 1) CSCs and mechanics are recognized as two key parameters involved in ovarian cancer metastasis, 2) a restrictive lack of knowledge is highlighted, and 3) recommendations are made for future research.

The review would fit well within the scope of the journal and in my opinion could be of interest to the journal’s readers. The writing is clear and concise and the spelling and grammar meet the required standard, although at various points in the manuscript the authors should have been more meticulous (see minor comments). Even though the topics of mechanics and stem cells have been reviewed previously in the context of ovarian cancer, the connection between the two in the context of metastasis could give a fresh perspective. To achieve this, I think the authors should consider the following:

Major comments:

1.             A vast issue with respect to cancer stem cells are the inconsistencies in the field. Nobody has actually isolated a well-defined cell population reproducibly. This makes it impossible to attribute properties, such as chemoresistance and invasiveness to one specific population, regardless of tumour type. The current manuscript also highlights this as the paragraph on ‘The roles of ovarian cancer stem like cells (CSCs) in metastasis’ exemplifies the inability to attribute a single set of markers to CSCs. Combined with the known cellular plasticity of marker expression, the overall view is that the concept of cancer stem cells may need to abandoned all together. To me, it does not suffice calling them cancer stem like cells in the manuscript. It still confers properties onto cells they may not have and vice versa. Overall, I think the authors should at least mention these issues with respect to CSCs in their review and give a critical view on the situation.’

We agree that it is difficult, if not impossible to isolate a single ‘cancer stem cell’ or ‘cancer stem like cell’ population given the plasticity of stem like cells and the many different markers associated with stem like properties. We understand that labeling cells as ‘stem like’ may give readers the impression that all ‘stem like’ cells have the same properties, however in this review we emphasize the importance of clearly defining the functions of all CSC subsets. That is to say, it is imperative that we look at expression of as many putative CSC markers as possible in our research to find out what stem like properties each set of markers does or does not have, and the special function of each CSC subset, if there are special functions. This would help all future researchers recognize the degree to which certain markers indicate stemness. We also suggest using at least two CSC markers in experiments requiring isolation of CSC populations to improve the robustness of any stem like cell claims. Finally, research on supposed CSCs should, and usually does, include evaluation of functional characteristics such as sphere forming ability, differentiation capacity, tumorigenicity, and chemoresistance, which should help to control for falsely labelling cells as stem like. Finally, we believe that the more comprehensive evaluation of CSC markers proposed may help to overcome the challenge of plasticity, allowing us to predict differentiation patterns and cell response to a given stimulus, the knowledge of which could then be used to direct therapy. A few sentences clarifying our stance on this issue have been incorporated in the ‘Challenges with incorporating CSCs in metastasis research’ section, as well as, the ‘The roles of ovarian cancer stem like cells in metastasis’ section.

2.       The coherence in the manuscript between the section on CSCs and mechanics is currently achieved mainly through one review paper, in which I was not able to find influence of CSC on mechanics and vice versa. I think the manuscript would be improved is this connection between the two topics could be strengthened, as now it reads almost as two separate manuscripts: one on CSCs and one on mechanics.

We have now included a section between the ovarian cancer stem cell section and the mechanical stimuli section, titled: ‘Nexus between cancer stem like cell phenotypes and mechanotransduction’. This section explicitly discusses the connection between CSCs and mechanical stimulation.

2.             Figure 1 contains some elements that deserve improvement. In panel A, it is not completely clear to me what the big circle represents. Is this the peritoneal cavity, with the arrows representing fluid flow or pressure and the small blobs representing cancer cells? I would also suggest changing Panel B, as in its current format, the figure implies cells are randomly dispersed in ECM. If the only goal with Panel B is to show the different cell types present, the legend suffices. Panel C seems to be an alternative representation of Panel D-F and could therefore be omitted. Otherwise, clarification is needed on what the streaks in the circle in C represent. In Panels D-F, detail on the causes of the mechanics would be helpful (i.e. what do the grey bars represent?). The legend of Figure 1F (line 151) reads: ‘… cause genotypic and phenotypic changes’. This is not represented in the figure and should therefore not be mentioned.

Figure 1 has been changed as per the reviewer comments. We intend to keep the schematic of diversity of cells in the TME, given the manuscript describes the interactions between different cell types, that are relevant to metastasis.

4.       Accuracy of referencing should be enhanced. For example, in the introductory paragraph, ovarian cancer statistics are discussed, but the references accompanying these numbers are not actual statistics papers. Overall, the authors reference other review papers in various instances, making it difficult for the reader to retrieve the actual scientific data that support the statements made.  Furthermore, in line 250, the authors refer to ref 55 ‘as we described previously’, I could not find any authors on that paper matching with the current authors. Please explain use of the word ‘we’.

The references and bibliography have been revised and we have included original articles, where possible. “We” referred to an article our lab published; it has now been correctly inserted. 

5.       The use of headings and sub-headings could be improved. 1) It is currently not clear where one section ends and the other begins: e.g. is ‘Interactions between ovarian cancer stem like cells and ovarian tumor microenvironment that modulate metastasis’ part of ‘The roles of ovarian cancer stem like cells (CSC) in metastasis’ or a separate paragraph? Is ‘Shear stress in ovarian cancer metastasis’ part of ‘Mechanical stimuli in ovarian tumor microenvironment and metastasis’ or not? Use of sub-headings would be appropriate if they sub-paragraphs. 2) Some of the headings are misleading in terms of content: e.g. the paragraph entitled ‘Tensile stress in ovarian cancers metastasis’ contains no information on ovarian cancer as ‘no such ovarian specific study has been performed’ and the paragraph on ‘Stress relaxation, creep compliance in ovarian cancers metastasis’  also continues into ‘There are currently no published studies investigating the effects of viscoelasticity on ovarian cancer at the time of this publication’.

We have revised the headings and subheadings. 

6.       The last paragraphs ‘In vivo studies on ovarian cancer metastasis’, ‘Challenges with incorporating CSCs in metastasis research’ and ‘Challenges with better in vitro and in vivo models to study mechanical stimuli’ are, in my opinion, not that well integrated in the manuscript. Now, these paragraphs come across as a sort of encore.

In lieu of moving the CSC challenges in metastasis research to the first half of the manuscript that discusses CSCs, we opted to add a transition paragraph before discussing the challenges. We believe that this change helps improve the flow from the mechanical stimuli section and the challenges section of the paper.

Minor comments: 

1.             Figure 1 B, the cell legend on the right: CAFs is spelled as CAF’s (see also my next comment) and Adipocytes is misspelled.

Thank you for the astute observation! CAFs have been written as CAFs throughout the manuscript. The spelling of adipocyte has been corrected. 

2.             Line 232-237: Three different variants of spelling CAFs (CAFs, CAF’s and CAFS) are used.

Thank you for the observation! We now refer to CAFs consistently as CAFs in the manuscript. 

3.             Line 224: Rephrase the sentence starting with ‘To address…’, it is not flowing well.

This sentence has been modified. 

4.             Line 244: ‘Elucidated’ seems an odd choice of words in this context. Suggest replacing with ‘Illustrated’.

We have replaced ‘elucidated’ with ‘illustrated’.

5.             Line 277: ‘…as well as hypoxia,’. It is not clear to me in what context hypoxia is mentioned here.

In this sentence, we meant to point to the role of hypoxia as a pro-inflammatory factor contributing to metastasis.  We have made it clearer in the revised text. 

6.             Line 340-343, this sentence needs referencing.

We have added the references to this sentence. 

7.       In line 349 ABCG2 is used, in line 463 ABCG2/BCRP, both are correct, but consistency in terms used would be good.

We have made ABCG2/BCRP consistent throughout the text.

8.       Line 409 and 428: ‘cancers’ should read ‘cancer’.

We have made this change.

9.       The formatting of the references is inconsistent, the majority of references are therefore lacking details (such as journal, volume, pages, some refs even lack a title).

We apologize for this irregularity! We experienced some errors during compilation of references. We have now re-inserted the references, and followed the journal guidelines for bibliography. 

10.   I would advise to reorganize the title. If CSCs are discussed before mechanics, I would also expect this to be reflected in the title. I would also replace ‘mechanical stimulation’ with ‘mechanics’ as the former may imply the discussion of how cells can be experimentally influenced.

We have rephrased the title to reflect the order of manuscript discussion topic and new wording for ‘mechanical stimulation’.

Round 2

Reviewer 2 Report

The authors have addressed my previous concerns. The addition of Figure 2 is very helpful in explaining the importance of mechanics.

Author Response

Thanks.

Reviewer 3 Report

The authors have addressed my comments and suggestions appropriately. However, I would like to suggest a final proof-read: the sentences in lines 213 and 305-306 are not flowing well. Also line 363 still refers to Fig 1F, which does not exist any longer in the revised version.

Author Response

We have addressed the minor changes suggested by reviewer 3.